# Discourses of change: The shift from infibulation to sunna circumcision among Somali and Sudanese migrants in Norway

R. Elise B. Johansen [ID]*

Norwegian Centre for Violence and Traumatic Stress Studies (NKVTS), Nydalen, Oslo, Norway

* r.e.johansen@nkvts.no

## Abstract

Somali and Sudanese transnational discourses on female genital cutting (FGC) center on a shift from infibulation to sunna circumcision, a change perceived to reduce health risks and accommodate religious teaching, yet this shift is far less extensive and substantial than its typical portrayal suggests. Based on data from interviews and focus group discussions with 95 migrants of Somali and Sudanese origin, in this paper, I explore these migrants' discourses of change and how and why they seem blurred and contradictory. Most participants described the ongoing abandonment of infibulation and uptake of sunna circumcision in terms of civilization, modernization and transition toward a more correct Islam; however, their perceptions of the anatomical extents and religious and cultural meanings of sunna circumcision appeared blurred and contradictory. We suggest that these blurred and contradictory perceptions of sunna circumcision enable the study's participants to maneuver in a context of opposing and changing social norms regarding FGC.

## Introduction

Several studies have identified a shift toward less severe types of female genital cutting (FGC), commonly referred to as sunna circumcision, in communities that have traditionally practiced infibulation [1–6]. This change is generally understood to reduce FGC-related health risks and to accommodate religious teaching due to a dominant perception that sunna circumcision is a minor and harmless procedure. Nevertheless, existing evidence suggests that this shift remains both less widespread and substantial than is commonly claimed [4, 7–9]. On the contrary may procedures described as sunna circumcision differ little from infibulation. Building further on this apparent contradiction, my aim in this paper is to explore how discourse of change regarding the shift from infibulation to sunna circumcision unfold among Somali and Sudanese migrants in Norway. Accordingly, the paper has two specific objectives: first, to describe how this change is perceived in terms of its anatomical extents and associated health consequences; second, to shed light on how these perceived changes are associated with changes in meaning making.

I start with a brief outline and comparison of the etic (external) and emic (internal to communities) FGC typologies as described by the World Health Organization (WHO), Somali,

**Data Availability Statement:** The data used for the analysis in this paper consists of transcribed interviews, which for anonymity reasons cannot be shared. However, relevant quotes and the minimal

data-set can be shared through request from postmottak@nkvts.no. The ethical review board NSD, now Sikt, requires that all data be anonymized, which is impossible to do with the interview transcript without it loosing its meaning. Therefore the ethical review board (NSD/SIKT) do not allow us to share transcripts, only a list of used quotes.

**Funding:** The author received no specific funding for this work.

**Competing interests:** The author have declared that no competing interests exists.

and Sudanese local communities. The WHO distinguishes four main types of FGC on the basis of anatomical extent and severity, with an increasing anatomical extent of tissue removal from I to III, though Type IV does not include tissue removal [10]. The emic categorization and terminology in Somalia and Sudan commonly identify two major types of FGC: infibulation, often referred to as pharaonic circumcision, and sunna circumcision, also referred to as *sunni* in Somali communities [11–13]. In terms of its anatomical extent, infibulation corresponds to WHO Type III. The term sunna circumcision, in contrast, is used to designate a wide variety of procedures, from a needle prick that corresponds to WHO Type IV to extensive removal and closure, similar to WHO Type III [4, 7, 14]. Based on a study among Somali migrants in Sweden, Wahlberg [14] has illustrated this difference in categorization represented in Fig 1 (S1 Fig, Fig 1):

Existing studies thus suggest that the emic differentiation between infibulation and sunna circumcision vary considerably. The broad definition of sunna circumcision has also been stratified into subtypes [8, 15, 16]. Table 1 outlines similarities and differences between the WHO and emic subcategories, as outlined in several studies, but mainly follows a recent study in Somaliland (the semi-independent Somali state) [7].

The categorization in Table 1 illustrates the ways in which infibulation and sunna circumcision are commonly differentiated in terms of both their amount and type of tissue removal and requisite stitches.

Worldwide, approximately 10% of the 200 million girls and women who have been subjected to FGC have undergone infibulation [17], yet this is the most common type of FGC in Somalia and Sudan. Of the 99% of women with FGC in Somalia, 64% report having undergone infibulation [11], and 77% of the 87% of Sudanese women who report having undergone FGC also report having undergone infibulation [18].

In Norway, the proportion of the 17,300 girls and women with FGC who have been subjected to infibulation constitutes approximately 50% [19]. This overrepresentation of infibulation is due to an immigration pattern; Somali women constitute about half of the total number of migrant girls and women assessed to have been subjected to FGC prior to immigration. Sudanese refugees and migrants constitute a considerably smaller proportion of female migrants affected by FGC, at approximately 3% [19]. However, due to the similarities in the

## Anatomical extent

No tissue removal → Tissue removal and closure

## WHO typology

| Type IV Pricking++ | Type I Clitoridectomy | Type II Excision | Type III Infibulation |
|---|---|---|---|

## Somali and Sudanese typology

| Sunna circumcision | Pharaonic |
|---|---|

**Fig 1.**

**Table 1. WHO classifications, juxtaposed with emic typology and terminology from Somalia and Sudan.**

| | |
|---|---|
| WHO Type I | *WHO*: Clitoridectomy. Defined as complete or partial removal of the (external part of) clitoris. |
| | *Emic*: Sunna circumcision, small circumcision. Defined as removal of the tip of the clitoral glans. |
| WHO type II | *WHO*: Excision. Partial or total removal of the clitoral glans and the labia minora, with or without excision of the labia majora. |
| | *Emic*: "Intermediate", "sunna2", "large sunna". Defined as removal of tissue from the clitoris and labia minora, followed by two to three stitches to stop bleeding and ensure a partial closure of the vaginal entrance. |
| WHO Type III | *WHO*: Infibulation. Defined as narrowing the vaginal opening with the creation of a covering seal by cutting and appositioning the labia minora or labia majora, with or without excision of the clitoral prepuce and glans. |
| | *Emic*: Pharaonic. Defined as removal of tissue from the clitoris and inner and outer labia followed by four to seven stitches, resulting in a miniscule vaginal opening. |
| WHO Type IV | *WHO*: All other harmful procedures to the female genitalia for nonmedical purposes, for example, pricking, piercing, incising, scraping or cauterization. |
| | *Emic*: Sunna. Defined as pricking, nicking or pinching the genitalia (commonly the clitoris or its foreskin) to ensure bleeding, without tissue removal. |

types of salient FGC, the cultural meanings and discourses of the shift from infibulation to sunna among both population groups, Sudanese are included in this paper.

Infibulation is traditionally associated with key cultural values that hinge on ideals and practices related to women's virginity and virtue and men's virility and sexual pleasure [20–22]. It is also associated with femininity, beauty and cleanliness [23]. Physical closure via infibulation is considered a form of a hymen, meant to create, protect and prove virginity [23, 24]. An intact infibulation at marriage is deemed evidence of a woman's virginity and virtue and thus her high moral standards [21, 25]. Failure to present an intact infibulation is, on the other hand, taken as evidence of premarital sexual encounters and thus immorality [5, 9, 21, 23, 26]. This is expected to lead to an immediate annulment of a marriage, causing the uttermost shame for the woman and her family. Infibulation is, however, also associated with male values, particularly virility and prowess, as men are expected to tear open the infibulated seal of skin with their penis to consummate their marriage [22]. Subsequently, infibulation, or rather the opening of an infibulation, is associated with sexual engagement and thus fertility [27]. Hence, the symbolic meaning of infibulation is intimately linked to the physical extent of the practice.

Nevertheless is infibulation also associated with a series of immediate and long-term health problems, many of which are well-known in practicing communities, including pain associated with urination, menstruation and childbirth. Painful and obstructed urination and menstruation refer to a slowed passage of blood and urine, creating pain from pressure that builds behind the vaginal seal [28]. Pain associated with sexual initiation and childbirth stems from the need to tear or cut open the infibulated seal of skin to create sufficient space for these activities [29, 30].

Additionally, the removal of all or part of the clitoral gland and foreskin that is common in infibulation is considered a measure that reduces a woman's sexual desire and therefore helps her maintain her virginity and virtue. This is built on the perception that the clitoris is both the site of sexual desire and perceived male values or characteristics, such as aggression and dominance [30, 31]. The clitoris is even commonly believed to grow to a penis-like size during puberty if it is not cut, resulting in an abhorrent organ that can obstruct sexual intercourse [30, 32]. This complex set of meanings that are associated with infibulation has been identified in studies of both Somalia and Sudan and among their respective migrant communities [20, 21, 23, 26].

As indicated above, the term sunna is generally used to describe all types of FGC that are not defined as infibulation. The term itself has an Arabic origin and is commonly used in Islam to designate religiously condoned practices. The practice of FGC is, however, controversial in Islam [33, 34]. While most Muslim communities do not practice FGC, most religious leaders in Somalia and Sudan support or accept the practice [7, 13]. However, the hadiths mentioning FGC, which supporters of the practice perceive to provide scriptural support for FGC, do not specify the correct anatomical extent of the procedure. They only offer admonitions not to "cut too deep" or "exaggerate" [7, 13, 33, 35]. Thus, in Islam, there is no clear or uniform understanding of FGC or of how it ultimately should be conducted.

Accordingly, to explore the discourses of change from infibulation to sunna circumcision, I draw on qualitative data from two studies that investigated the changing perceptions of FGC among Somali and Sudanese migrants in Norway.

## Materials and methods

Data collection included in-depth interviews (IDIs) and focus group discussions (FGDs) with 95 Somali and Sudanese women and men. This was carried out between the fall of 2017 and the fall of 2018 in two parallel studies. Fifty participants took part in the IDI, and 45 participants took part in a total of eight FGDs with three to nine participants in each. The majority were of Somali origin (n = 87) and a minority were of Sudanese origin (n = 8), corresponding closely to their proportions of the FGC-affected communities in Norway, 53% and 3%, respectively.

The first study included IDIs and FGDs with 72 Somali women and men, all of whom resided in or close to a medium-sized Norwegian town. The second study included IDIs with 23 Somali (no = 15) and Sudanese (no = 8) women who had settled in eight different towns and villages across Norway.

All FGDs and most IDIs (all in the first study and seven in the second study (n = 79) were carried out by seven research assistants matching the participants in terms of ethnicity or national background and gender (male/female). The research assistants included three Somali and one Sudanese woman and three Somali men. They used Somali, Arabic, Norwegian, English, or a mixture of these languages. FGDs were, with one exception, segregated by gender and age group, as this was expected to facilitate discussion through more shared life situations and social expectations. For example, do the sexual norms of virginity imply that unmarried persons, particularly women, would likely feel uncomfortable discussing such topics with their mother's generation. Other experiences, such as childbirth or marital relationships, were also less likely to be shared experiences that could generate free and open discussions across age groups. One FGD however, mixed ages, with two young and three adult men. Division into age group for the FGD was based on the participants' self-identification, with the demarcation between "youth" or "adult" generally corresponded with marital status. The IDIs and FGDs were tape recorded, transcribed verbatim, and then translated into Norwegian or English by the research assistants before they shared them with the author. The remaining interviews (n = 16) were conducted by the author, an ethnic Norwegian woman, and were conducted in Norwegian, English, Kiswahili or Somali, the last with the help of an interpreter. Most interviews lasted between one and three hours, were open and informal, and were conducted at a place chosen by the informant.

In both studies, we asked for background information including childhood and upbringing, experiences of pressure for or against FGC, perceived differences between country of origin and Norway, perceived relevance of FGC to the sense of identity and belonging, and perceptions of the change in FGC practices. The first study included additional questions on

perceived motivation for FGC, decision-making, health risks, legal issues and ultimate relevance of FGC to marriage (S1 and S2 Files). The second study included additional inquiries about their first encounters with FGC as a practice, critical discourses about the practice, participants' social networks and their reflections on FGC regarding their eventual daughters (S3 File).

To obtain a broader impression of such discourse, additional data were collected through ethnographic fieldwork, including in-depth interviews, key-informant interviews, informal conversations, group discussions and participant observation. Most of this data-collection was carried out in Norway, whereas a small part was collected during a one-month stay in Kenya in March 2019. This included the following activities:

In the first study, the research team (author and research assistants) arranged three community meetings for potential study participants prior to the study, involving approximately 30 people. After the completion of data collection, we (the author and two of the research assistants) arranged two FGDs with community members (one with 9 young women and one with 10 adult men). Here, I presented a preliminary analysis to determine how our findings resonated with their perspectives and to give the participants an opportunity to correct and expand our analytical perspectives.

The author also conducted in-depth interviews and informal conversations with other Somali and Sudanese women and men. This included onward migrants—Somali and Sudanese migrants who had left Norway to resettle in other countries of migration (UK and Canada)—and return migrants—Somali and Sudanese migrants who had permanently left Norway to resettle in their countries of origin including Sudan, Somalia and Kenya. Some of these interviews were conducted over the phone. The author also gained additional insight through participant observations in various settings, including FGC workshops and seminars on women's health and FGC with Somali and Sudanese participants. Interviews with key informants included persons working in nongovernmental organizations (NGOs) (mostly of Somali origin) and government agencies dealing with FGC in transnational contexts (of British, Norwegian and Somali origin).

Findings from the ethnographic fieldwork is, with one exception (an interview of a Somali midwife in Kenya), not directly referred to in the paper, but was used as background information during the analysis to improve and check the analysis and its validity.

Some aspects of the discourses regarding the change from infibulation to sunna circumcision have been alluded to in two formerly peer-reviewed and published papers, using data from the same studies [5, 36]. Whereas the former papers presented an overview of all themes identified during data-analysis, the aim of this paper was to make an in-depth analysis of one of the major findings, that is, the discourse of change from infibulation to sunna circumcision. For the purpose of this paper, I thus re-read and reanalyzed all transcribed data from both studies to identify all data relevant for the discourse on change of type. Thereafter I systematized the data into major themes and subthemes and used them to organize this paper. Thus, this paper includes considerably broader empirical evidence on the change of type, and a more nuanced and elaborated analysis than previous papers.

## Research participants

The 95 research participants included 65 women and 30 men. Participants were between 16 and 59 years of age. All research participants were first-generation immigrants, born and raised in Somalia, Sudan or a neighboring country, most commonly Kenya, Ethiopia, the United Arab Emirates or Yemen. Some had arrived in Norway in their early childhood, others during their youth, but most as adults. Their times of residence in Norway ranged from six months to thirty years.

Most young participants claimed to be students at a high school or university, whereas few of the adults provided reliable information on their current engagement in education or employment. Educational backgrounds varied extensively, from one year of Koranic School to a PhD. Overall, the Sudanese participants had higher levels of education than the Somali participants, with several of the former having a postgraduate university education.

Most female participants (n = 65) shared information about their personal FGC status (n = 57). Of these, the majority revealed they had undergone FGC (n = 50), mostly infibulation (n = 37). Five women claimed to have undergone sunna circumcision, of which four reported extensive vaginal closure. One woman was unclear about type, and not sure what type of FGC she had, and six participants claimed to have no FGC.

### Ethical considerations

The studies followed the ethical considerations required by the Norwegian Centre for Research Data (NSD), which confirmed that the study was in line with the personal data act (reference no. 55641). All the potential participants were provided with a written and oral presentation of the study and the implications of their participation in their selected language (Norwegian, English, Arabic or Somali). All participants gave their informed consent orally (tape recorded) or written according to their choice. In Norway, parental approval is not required for anyone over the age of 16 years; thus, this was not needed. To ensure anonymity and reduce the risk of recognition, details about the study's participants are left deliberately vague.

### Reflexivity

As indicated, most of the interviews were conducted by a research assistant with the same ethnic and/or national background as each study participant, and the author, an ethnic Norwegian, collected some interviews and all ethnographic data. Similarities and differences in personal factors, such as ethnicity, age, education, language proficiency and (perceived) FGC status, between the interviewers and interviewees are likely to affect access, trust and thus data quality. While there were no systematic differences found between the data collected by the author and those obtained by the research assistants in this study, such differences may occur, as described in an earlier paper [37].

Although there were many similarities between the two studies that the data in this paper are drawn from, the second study did not address the conceptualization of sunna circumcision unless it was raised by its participants. Thus, most of the quotes in this paper are from the first study. Quotes and stories drawn from the second study are marked with a star (*). Quotes and stories from Somali participants marked with a star are from the interviews conducted by the author, whereas the rest are from interviews conducted by the research assistants.

### Findings: Discourses and perceptions of a shift from infibulation to sunna FGC

All the participants expressed a negative attitude toward infibulation and believed this negative attitude to be the dominant social norm in their local community in Norway and a growing social norm in their countries of origin. One example of this is a statement by an adult man during a discussion on FGC abandonment in Somalia: "*I think people have stopped practicing infibulation in the large cities. They use sunna circumcision instead, because it is not dangerous*" (FGD). Most study participants, however, described the shift from infibulation to sunna circumcision in terms of FGC abandonment. When talking generally about FGC, participants seemed in general to refer to infibulation. When we wanted their views on sunna circumcision on the other hand, this had to be specified. This often became evident indirectly, as in the

following example, where a Somali woman recalled how she had successfully discouraged the circumcision of her to nieces in Somalia, saying that: "*My sister has not circumcised her two daughters.*" Later in the interview, however, she specified that she had succeeded in saving them from infibulation as *"Or, yes, she had them undergo sunna, Type I and II"* (IDI). * This is particularly interesting as this woman had been engaged in work against FGC in Norway, and thus had more in-depth knowledge and reflection that the average migrant.

## Negotiating the abandonment of infibulation

Most study participants grounded their resistance against infibulation in a concern about its negative health consequences. Furthermore, all the health consequences outlined by participants were related to extensive vaginal closure, such as painful and obstructed menstruation, urination, sexual initiation, and childbirth.

Many participants also suggested that these health complications designated infibulation contrary to religion. In contrast, no participants considered sunna circumcision to be haram, i.e. religiously forbidden. There was thus a perception of a religiously right or wrong way to conduct FGC, as formulated by a young Somali woman: "*I am not saying the FGC is haram [sinful], but the way people do it is wrong. (. . .) They don't do it the way it is described by religion*" (IDI). This distinction between a religiously wrong or right way of FGC was also framed in terms of a contrast between tradition and culture, which are manmade and thus changeable, and the eternal values of religion, as expressed by an adult Somali woman:

> "*Culture is wrong. It is something humans have invented. Religion, in contrast, comes from God, and religion says that FGC [infibulation] is forbidden. This I heard when I grew up in Somalia, when the Koranic School said that FGC is not good, it is only culture*" (IDI).

Participants also commonly dismissed infibulation as an outdated and dying practice, which only occurred in rural areas among uneducated and impoverished individuals. This they often contrasted with the practice of educated migrants with an urban background, described as civilized and informed, as in this claim by a middle-aged Somali man:

> "*Somalis in Norway are more civilized than those in the country of origin. They have got information, and some also have a higher education from before. Those who did pharaonic [FGC] lived in the countryside and did not get the opportunity to move to Norway*" (IDI).

Many participants suggested that infibulation had been imported from Egypt, with some inferring this from the terminological parallel between pharaonic FGC and the Egyptian pharaohs, others through a reinterpretation of religious texts (Old testament Exodus 1:15–22, Koran (Quran): 28:1–6), such as in this elaboration by an adult Somali man:

> "*(. . .) to limit the birth of babies, he [the Egyptian king] decided to control women's sexual desire or behavior and decided, together with the minister of health, that they should start cutting away female genitalia and stitch it together. Later, this practice was spread to other groups or states around Egypt, with which Egypt had financial, cultural or political connections. After several thousand years, it came to Somalia and became a part of our culture*" (FGD).

Similar perceptions of the Egyptian origin of infibulation have been recorded elsewhere [32, 35].

Some of the Somali men used this perception of an Egyptian origin to distance themselves from the practice, framing it as an imported and foreign practice and thus not a genuine Somali tradition. For most Somali women, in contrast, the perceived pharaonic origin did not lead to a similar distancing. None of the Sudanese participants mentioned an Egyptian origin, though it has been a common perception in Sudan as well [38]. For most participants, however, the link to a pharaonic origin placed the practice in the past, and this distant historical origin was used as evidence that the practice is outdated.

The historic origins of infibulation were often contrasted with the newness of sunna circumcision, which some of the participants dated to the 1990s, as a Somali man in an FGD: "*Pharaonic [FGC] is ancient, from the Egyptian pharaohs, whereas sunna came about in the early nineties*".

Generally, then, all participants were negative toward infibulation, and most believed the practice to be waning. Most participants, particularly the men, further dissociated themselves from the practice by defining it as distant in terms of time, place (rural areas) and degree of civilization (uneducated, poor, and ignorant).

## Perceptions of sunna circumcision

While describing sunna circumcision, most participants used infibulation as a reference. Participants generally described it as a modern procedure, common among educated and urban cosmopolitans, religiously promoted or approved. They also claimed it to be less extensive and harmful than infibulation, though rarely provided any concrete descriptions of its anatomical extent.

Concrete depictions of the physical extent of sunna circumcision, however, varied extensively, both between and among individuals. It was described as anything from a minor pricking to allow some blood to flow to an extensive removal, commonly followed by two or three stitches. Many participants explained this variation as a distinction between an ideal and a real way to perform sunna circumcision. Ideal sunna circumcision was considered a type of FGC that is encouraged, supported or accepted in Islam and was generally described as minor in terms of its physical infringement. Most participants grounded their conception of a religiously correct form of FGC as a procedure with minimal cutting in their broader religious values of avoiding bodily harm. In contrast to ideal sunna, many participants described real sunna circumcision, i.e. as they believed it to take place in reality, as a quite extensive procedure that could include both extensive tissue removal and vaginal closure.

Nevertheless, most commonly, the participants contrasted both ideal and real sunna circumcision with infibulation by referring to vaginal closure. In contrast to the tight vaginal introitus associated with infibulation, all participants described sunna circumcision in terms of a lack of, or less extensive, closure. One example was a Somali woman in her late fifties: "*Sunna has an opening, with less cutting. That's the difference, stitched versus open*" (IDI). However, this distinction was often vague and sometimes even contradictory, as indicated in the statement of a Somali woman in her forties:

"*Sunni is no cutting or stiches. You can do two stitches, so it's very little, almost no cutting. Sunni is a small procedure; there is less pain and only two stitches, so the woman will have less pain during sex, periods and childbirth*" (IDI).

Many participants shared this view of sunna circumcision as involving some stitches and hence some vaginal closure. How extensive such closure was perceived to be, however, varied. One Somali woman, for example, distinguished the closure of infibulation from sunna

circumcision by pointing at the two ends of a pencil, the writing end indicating the closure in infibulation and the rubber end that in sunna circumcision (*). Following her indication it would imply that in infibulation the vaginal introitus would be only 3–4 mm. in diameter, whereas the vaginal opening left in sunna circumcision could have diameter of 2,5 cm. Another Somali woman claimed that sunna circumcision entailed closure until the urethra, which she illustrated with a drawing (*). However, the most common way of describing the extent of closure in sunna circumcision was a reference to the number of stitches, commonly two or three. No participants specified how this differed from the number of stitches in infibulation. However, as indicated in Table 1, the common number of stitches in infibulation is between four and seven. Most participants did not provide any clear or shared idea of the extent of the closure that the stitches in sunna circumcision would form. The only exception was a Somali midwife I interviewed during my stay in Kenya. She was working at a health center catering to Somali women and claimed that since FGC is performed on small girls with equivalently small genitalia, two to three stitches cause significant vaginal closure. She also claimed that she had never seen a sunna circumcision without closures sufficient to hinder vaginal intercourse. She insisted that circumcision (both infibulation and sunna) has to include closure that effectively makes sexual intercourse impossible without tearing or cutting parts of the infibulated seal. This is in line with findings from a recent study from Somaliland [7].

Some of the participants, however, expressed concerns due to the apparent lack of any substantial difference between infibulation and real sunna circumcision, e.g., these reflections of a young Somali woman:

*"I have heard that sunna is a lighter type of cut, but it is not that different from the other type. I feel that nobody follows what the prophet says. Sunna has a larger hole than pharaonic [FGC]. When I left Somalia eight years ago, pharaonic was the most common. But, you could not really see the difference, as girls were cut almost in the same way, whether it was pharaonic or sunna. That means it is all closed and you cannot see any major differences. Even the girls themselves or their parents do not know how to differentiate between the two types. I don't say that circumcision is sinful, but the way people do it is wrong. They don't do it the way it is prescribed by religion"* (IDI).

In these elaborations, this participant suggests a lack of clarity concerning any difference between infibulation and sunna, both among parents and providers. As she was not a healthcare provider, the basis of her observations is not clear. Nevertheless, as it is common in Somalia for young girls to show each other their circumcision and discuss the size of their vaginal openings [30, 39]. Her observations could have stemmed from such experiences. Nevertheless, she seemed to conclude that severe types of FGC are religiously sinful, whether referred to as infibulation or sunna circumcision, which she distinguishes from an ideal form of sunna circumcision prescribed by religion.

The amount of tissue removal was the other major distinction made between infibulation and sunna circumcision. This was generally described rather vaguely, as in this description by a Somali woman in her early thirties: *"(. . .) you cut (slice/cut with a scissor) some part of the clitoris. You don't remove everything, but you cut off some part. It is a professional doing it. You cut some of the upper layer, maybe a part of clitoris, it's so little"* (IDI). She then substantiated her description by arguing against closure via infibulation, which she described as *"stitching up what God has left open"*. Nevertheless, while describing her own circumcision as the sunna type, her claim that she had to *"remove the stitches"* during childbirth suggests that her sunna circumcision also included vaginal closure.

Some women related differences concerning the extent of cutting in sunna circumcision to anatomical variations. In an FGD for adult women, one participant argued *"Bodies differ. [In sunna circumcision] they cut what has grown. Skin that grows"*. One of the other participants substantiated this argument, responding that *"Sunna implies less circumcision [than infibulation]. There is a difference in the thickness of the skin in women's genitals. For some women, a clitoris can grow and need to be cut. I suppose this is sunna. But, the labia are still intact."* Such suggestions of a difference in cutting based on anatomical variations among girls have not been previously reported in Sudan or Somalia, although they have been described in other settings [40, 41].

A few of the participants described ideal sunna circumcision as pricking or nicking the genitalia to draw some blood without removing any tissue, e.g., two Somali women, one in her forties and the other in her fifties, who stated, respectively: "*sunna is normal, just a tiny prick to cause some bleeding*" (IDI) and "*No cutting, no stitching and no tying of legs*" (IDI*).

Male participants were generally even less specific than women in their outline of the anatomical extent of sunna circumcision. One example is the description by a Somali man in his late twenties: *"Sunna circumcision is common and normal. The girl doesn't lose any part of her body" (*IDI*)*. An even younger man described sunna as the removal of *"very little tissue" (*IDI*)*. For this reason, he considered it significantly less dangerous than pharaonic FGD, although not necessarily totally harmless.

Overall, while there were extensive variations in the depictions of the anatomical extent of sunna circumcision, there was a general agreement that the procedure was significantly less extensive than infibulation, included less or no vaginal closure and often entailed less tissue removal.

Deemed less physically infringing than infibulation, sunna circumcision was commonly depicted as harmless. Its lack of health complications was commonly explained by a larger vaginal orifice. Even those who depicted sunna circumcision as including some vaginal closure believed this closure to be less tight and hence not obstructive of women's natural functions of urination, menstruation or sexual relations; at least, not to the same extent. This became clear when we probed for the perceived differences between the health problems caused by sunna and infibulation. A Somali woman in her forties, for example, described, during an interview, how sunna was the removal of "*Just a little piece. The girls don't get so much damage*". A Somali man in his early thirties claimed that "*Sunna circumcision is less dangerous and good for the girl's health. Sunna is easier and less dangerous. I support sunna*" (IDI). A Somali man under 20 years of age similarly claimed, during an interview, "*Sunni is to cut a small piece of the girl's genitalia. It is accepted religiously. It is the type where you just cut a little, and it is less dangerous than the pharaonic [FGC]. I have not heard that sunni circumcision gives any health risks or affects her sexual drive*".

These quotes illustrate that while most Somali participants presented sunna circumcision as harmless, they simultaneously perceived the practice to typically be physically extensive. Although some participants revealed that sunna circumcision could cause some harm, this was considered rather less severe than the harm caused by infibulation and thus better. Accordingly, when participants suggested that sunna circumcision is *"good for the girl's health"*, they seem once again to use infibulation as a reference, i.e. that sunna is better for their health than infibulation, and not that it is good for female health per se.

Only one participant mentioned concrete experiences with health complications caused by sunna circumcision. This was a Sudanese woman with infibulation, whose youngest sister had undergone sunna circumcision, causing her immediate and long-term health problems*. A few of the men also suggested that sunna circumcision might cause some harm, although they did not report any personal experiences. This included a man of approximately 50 years of age

who, after studying the subject in preparation for his planned interview had concluded that sunna destroys an important body part for women. Another Somali man, approximately 30 years younger, concluded that while he did not consider sunna circumcision dangerous, he thought it could cause some harm.

Furthermore, there was an almost uniform view that sunna circumcision has no negative consequences for women's sexuality. This was often specified in terms of the absence of pain during sexual initiation, which is common with infibulation. This focus on the absence of pain, however, disregards other aspects of sexual encounters, including sexual desire, pleasure and satisfaction for women. This was striking; most participants described sunna circumcision, in both its ideal and real form, to include removal of clitoral tissue, and both Somali and Sudanese migrants shared the perception that the clitoris is the physical site for women's sexual desire and pleasure [9, 26, 30]. Hence, a depiction of sunna circumcision as lacking consequences for women's sexuality may reflect a limited appreciation of women's sexual desire and pleasure. It may even reflect a generally negative perception of female sexual desire and pleasure due to its perceived link to sexual immorality, as other studies have found [26, 30, 34]. This is linked to the traditional perception that a lack of FGC is associated with promiscuity, not sexual pleasure [5, 20, 21, 23]. This interpretation is affirmed by many of the stories of participants, which described the change from infibulation to sunna circumcision as a means to reduce any associated health risks while maintaining the meanings that associate FGC with sexual control and virtue. A Sudanese woman in her fifties recalled, during an interview, how her sister in Sudan had tried to convince her to have her daughter cut by referring to a medical doctor's argument that sunna is "*a small thing, just to minimize lust*"*. A Somali woman in her forties was told that sunna circumcision "*it helps, because when a girl has her clitoris, she will desire*" (IDI). A Somali woman in her thirties, claiming to have been subjected to sunna circumcision with stitches, after which her legs were bound, laughingly expressed how she had told Somali friends that she has no FGC to see their reactions: "*We friends, we touch each other, right. (. . .) There is a girl who touched my breasts and bum—they laugh and think I get so horny since I am not stitched*".* Another allusion to the notion that the removal of the clitoris is a way to reduce sexual energy was articulated by a few participants who likened FGC to the castration of male camels; both procedures are believed to be calming.

However, a few participants, mainly young women, expressed concerns that sunna circumcision could also have negative sexual effects on women, such as a Somali woman of approximately 20 years of age: "*I don't think there are any sexual consequences of circumcision if one is open. When it comes to sunna, it depends how much is removed*" (IDI).

There thus seemed to be an apparent inconsistency in the participants' views regarding the motivation for and effect of sunna circumcision, i.e., sunna circumcision was perceived to have no effect on women's sexuality while simultaneously reducing women's sexual desire.

This shifts our focus back to the ideal of virginity, which is a central aspect of infibulation. Sunna circumcision in its ideal form should not include (extensive) vaginal closure and thus may not function of constructing, protecting and proving virginity as infibulation is considered to do. Thus, below, we explore how the participants expected the transition from infibulation to ideal sunna circumcision to affect virginity ideals and ideas of marriageability.

Most participants claimed that FGC was not important for marriageability in the diaspora. This claim implied that no Somali or Sudanese man living in Europe would demand that his wife be infibulated. Furthermore, several participants suggested that many men preferred their wives not to be infibulated to avoid painful sexual initiation. Meanwhile, however, there was a common perception that unmarried women without infibulation, including women who had undergone defibulation, would be deemed promiscuous and immoral. Defibulation is a minor surgery to open the infibulation seal to expose the vaginal and urethral openings and may thus

ease any health consequences caused by closure [42]. The general view of such procedures as socially unacceptable if conducted premaritally could be considered a contradiction to the uniform consensus against infibulation.

Thus, while participants grounded the change from infibulation to sunna circumcision in a concerns over health, sexual problems, and the perceived backwardness of infibulation, some of the cultural values associated with FGC, particularly virginity and virtue, seemed to obstruct the speed and extent of this change. In the following section, I thus elaborate on the meaning making that dominates the participants' discourses of the change from infibulation to sunna circumcision.

## Meaning making in the change from infibulation to sunna circumcision

Study participants generally viewed the change from infibulation to sunna circumcision as a change not only in physical extent but also in meaning. Participants commonly discussed this change along three dimensions: as a change from culture to religion, from external to internal control of female sexuality and from old-fashion to modern.

### From tradition to religion

Many informants explained their own shift from supporting infibulation to supporting or accepting sunna circumcision as a part of their religious education and awakening. They had come to view infibulation as not required by Islam and even contradictory to their faith, i.e., as haram. Islamic jurisprudence issues the term haram to refer to any act forbidden by God, typically in the Quran. While infibulation is not mentioned in the Quran, the participants' perceptions that it is haram were grounded in its severe health consequences. Regarding their religious views on sunna circumcision, many participants were aware of various hadiths and of some of the controversies in terms of their interpretations. Furthermore, while the Quran does not mention FGC, some participants thought that it does and thus deemed the practice obligatory.

Many participants framed the abandonment of infibulation as a change from tradition and culture, which are dictated by humans, and thus could be defied. In contrast, as religion was perceived as God-given, those who perceived sunna circumcision as a religious practice, thought this could not be challenged. As indicated by an adult Somali man in an FGD: "*There are two types of FGC in Somalia; the one, pharaonic is cultural, and the other, sunna, is religious*". A Somali woman expressed a similar view during an interview: "*Before, with infibulation, this was done to control women [sexually]. But, nowadays, with sunni, it is done for religious reasons*".

Whereas the use of the sunna term to designate a type of FGC provides the practice with a religious grounding or framing [43], the link between FGC and Islam was differently perceived by the participants. Some saw the practice as obligatory, others as beneficial and still others as merely accepted. These views were not always based on a clear or strong conviction but rather on an insecure deduction, as expressed by an adult Somali man during a FGD: "*Sunna is, as the name indicates, associated with religion. At least, they say it is more in accordance with religion*".

The overall impression was thus that while most participants reinterpreted infibulation as contrary to Islamic teaching, they perceived sunna circumcision to be a replacement that was religiously encouraged, supported or at least accepted.

### From outer to inner sexual control

The second aspect of changes in meaning relates to the perceived sexual significance of FGC. Whereas the meaning of infibulation is intimately tied to closure, constituting a physical

hindrance to and evidence of a lack of sexual engagement, the reduction or absence of closure said to be characteristic of ideal sunna circumcision suggests a change in meaning. Often, participants described a transition from infibulation to sunna circumcision as a change from outer to inner sexual control. Thus, to replace the physical closure endured through infibulation, they emphasized the importance of inner control, which could be ensured through improved parenting, religious education and reduced sexual desire through sunna circumcision. One example was an explanation provided by a Somali man in his thirties:

> *"In our parents' time, they thought to circumcise and stich the labia together, so that girls could not have sexual intercourse. So, they thought that this was the solution to stop girls from having sex. But, they didn't think about giving the girls information and knowledge about how not to do it [premarital sex]. You see? They just thought about closing it [physically] to prevent sexual activity"* (IDI).

References to clitoral removal as a measure to reduce women's sexual urges, and thus their sexual self-control, were also often alluded to. One example is the recollections of a Sudanese woman in her fifties about her sister's suggestion to at least subject her daughters to sunna circumcision:

> *"Your daughters are still young, so you don't know how girls would survive the teenage period; are we going to run behind them all the time? It is just a small thing! Just to minimize their sexual desires so they won't make troubles. Girls also start to wear jeans, and friction can stimulate them too"* (IDI)*.

While statements such as this were commonly presented in terms of arguments from others, the participants rarely disclaimed their validity. In general, discourses of the change from infibulation to sunna circumcision were intertwined with a discursive change towards increased privatization and internalization of sexual control. In contrast to infibulation, which is performed, checked, and broken by others, sunna circumcision in its ideal form was perceived to be a procedure that contributes to sexual virtue by reducing sexual desire. This discursive shift seems to have begun in parallel with an increased focus on religious teaching of sexual morality.

These blurred discourses of a change from outer to inner sexual control trough a change of type of FGC, were counteracted by the limited extent to which this was perceived to have taken place in practice. That is, sunna as it was perceived to be commonly practiced in reality, often included sufficient closure so as to ensure also external control. Furthermore, was there a simultaneous discourse of increasing other forms of external control, particularly in the form of pressure to follow a modest dress code and socialization- and movement pattern. This suggests that the discourse of transition is complicated and remains primarily at a discursive and ideal level.

## From old-fashioned to modern

The third major change in meaning is the discursive framing of the shift in the type of FGC as a civilizing process. Participants commonly associated infibulation with the past, with rural areas, and with a lack of education, as indicated by a Somali man in his thirties:

> *"People in Somalia are now beginning to leave this bad culture behind. They are enlightened. They know how it is not good for girls when they marry, for their husbands, or when they are giving birth or for their sexual life. So, people who have information and are enlightened are*

*abandoning this. But, people in the rural areas, those who we call baadiya, they continue to practice the pharaonic type"* (IDI).

Similar perceptions were expressed by another Somali man:

"*If we talk about the old times, then pharaonic circumcision was common among Somalis. But, nowadays, it is different. It seems people have become more civilized now, particularly in large cities. They use sunni circumcision. I think we should continue to practice circumcision, but not the pharaonic type*" (IDI).

Many participants shared similar reflections by framing the change in FGC type as a result of modernization, education, civilization, and migration. Thus the shift was described as a result of new insights gained through education, religious enlightenment, and exposure to international impulses through travel or discussions with returned or visiting migrants.

## Discussion

In this paper, I have explored various aspects of the discourses of the change from infibulation to sunna circumcision by revealing how it was reflected on by Somali and Sudanese migrants in Norway. My purpose was to investigate what is at stake and what the change is about, both in terms of its physical extents and meaning makings. The perceived changes were presented as a reduction in the physical extents of FGC, mainly in terms of vaginal closure. This change was believed to reduce or escape FGC related negative health consequences. Simultaneously, I identified changes in discursive meaning making along three lines, from culture and tradition to religion, from outer to inner sexual control, and from old-fashioned to modern.

The framing of the change from infibulation to sunna circumcision as a change from tradition and culture to religion was grounded in both concerns for health and terminology. The concern for health was related to a perception of infibulation as a practice that causes severe health consequences and is thus contrary to Islamic values. In contrast, sunna circumcision was considered minor and harmless and therefore at least religiously acceptable. The difference in terminology, from pharaonic to sunna circumcision, indicates a change from an old historical and, to some extent, foreign practice to a practice condoned by Islam. There also seemed to be indirect support for sunna because it was perceived to reduce sexual desire and to thereby contribute to sexual morality in women. The linking of sunna circumcision to religion was also underpinned by experiences with religious leaders who support the practices, which have also been documented in Norway [39]

Furthermore, most participants deemed sunna circumcision harmless, with no negative consequences for women's sexuality, regardless of whether they spoke about ideal or real sunna circumcision. On the other hand, many simultaneously expressed views that suggested a belief that the removal of clitoral tissue in sunna circumcision reduces women's sexual desire. However, this seemed not to be of great concern. Rather, some of the statements could suggest that some participants perceived this effect as a positive contribution to secure female sexual morality in the absence of infibulation. Moreover, the common perception that real sunna includes physical closure sufficient to hinder vaginal intercourse suggests a continuation of cultural values, i.e., of the need for physical closure to ensure virginity.

The perception of the change from infibulation to sunna circumcision in Somalia and Sudan in terms of civilization was associated with ideas of urbanization, education, international migration and religious training. However, these perceptions are only minimally supported by available statistics. If infibulation was only practiced in rural areas, this would still

constitute 65% of the population in Sudan and 50% of the population in Somalia. Furthermore, the most recent prevalence in Somalia of 99% offers limited room for variation between groups. In terms of the changes in the type of FGC, the common claims of abandonment of infibulation in urban areas are not confirmed. The proportion of women in Somalia who report having undergone infibulation varies minimally between rural (65%), nomadic (62%) and urban (63%) communities [11]. Education, in contrast, was associated with a lower prevalence of infibulation, but 50% of highly educated women had still undergone Type III. Thus, while there is statistical support for a reduced prevalence of FGC and for less extensive types of FGC in urban and educated families, these differences are significantly less than was commonly claimed by the research participants.

The findings reported in this paper suggest a higher acceptance or support of sunna circumcision than other studies on diaspora populations have indicated [3, 44]. This divergence is probably related to this study's method (qualitative versus quantitative), focus (accept and meaning versus support), and place. The aspect of place relates to the site of the first study, which was conducted in a small town with a rather conservative and tightly knit Somali community. Thus, these participants may be more conservative than an average Somali migrant in Norway and certainly more conservative than the Somali migrants in Oslo who were included in the study by Abdi Gele [3]. The second study, on the other hand, was probably biased toward women eager to abandon all forms of FGC, as it included women who were specifically targeted for their advocacy against the practice and their interethnic marriages [36].

The overall impression of the discourses of the change in question are thus that they are blurred, lacking any agreement or clarity about what changes are happening and should happen in terms of the anatomical extents and meaning makings of FGC. How then, should this blurred discourse of change be understood?

To answer this, I will draw further on insights from a paper that used data from the second study to explore how migrant Sudanese and Somali women maneuver between contradictory social norms [36]. On the one hand, they had to relate to the traditional social norms of FGC, infibulation in particular, as the ultimate measures of women's morality. On the other hand, they had to relate to the international social norms that define FGC, and infibulation in particular, as the ultimate expression of female oppression and a violation of human rights. The majority of the participants expressed a sense of ambivalence and of being torn between these sets of norms, similar to what has been found in other studies [39, 45]. Thus, on the basis of my reanalysis of the data from the two studies, I suggest that the blurred perceptions and discourses of sunna circumcision presented in this paper constitute a potential way to facilitate women's and men's balancing act between traditional and international sets of norms. By presenting sunna circumcision as a minor and harmless procedure, Somali and Sudanese migrants can refute the international classification of FGC as a violation of human rights. The term sunna and the perception that it is a minor procedure may also be useful for building bridges with wider Muslim communities. As most Muslims in Norway and internationally do not have a tradition of FGC and thus tend to refute its Islamic roots, referring to the practice as sunna and minimizing its physical extent can help build bridges with other Muslim communities [35, 45]. This may be particularly important for diaspora communities, as Somali and Sudanese migrants, like many other migrant groups, experience a sense of religious awakening and engagement amid diaspora. This has also been related to the desire to belong to a broader community, which can become even more important than ethnic or national identity in a diaspora [46–48]. Additionally, vis a vis international norms, a shift from traditional to religious underpinnings of the practice may receive some support, as religion has a stronger legitimacy than culture and tradition in all countries, including their national and international policies, e.g., in clauses regarding "the freedom of religion" [10]. Nevertheless, can the understanding

of sunna as a general term designating a wide variety of practices, facilitate agreement with family members and other conservative forces in migrants' country of origin. This makes it possible for a person who wants only a tiny prick and a person who supports extensive tissue removal and closure to agree, as they may both agree to sunna circumcision. I thus suggest that the blurred discourses of sunna circumcision may constitute a way for a careful negotiation of change to conciliate the highly politicized and sensitive discourses in countries of origin, Norway and in international fora.

## Concluding remarks

In this paper, I have explored an ongoing international discourse on the change in FGC from infibulation to sunna circumcision that is dominant in Somali and Sudanese communities in Norway. I found that the discourse on the abandonment of infibulation was largely formulated in terms of avoiding health risks and improving religiosity, inner sexual control and civilization. In contrast, most participants depicted sunna circumcision as a modern, minor and harmless procedure in line with religious teaching. In theory, the transition from infibulation to sunna circumcision was depicted as a shift from vaginal closure to openness, associated with a change in meaning from tradition to religion, from outer to inner sexual control and from old-fashioned to modern. However, in reality, both the anatomical extents and associated cultural meanings of FGC were blurred. I thus suggest that the blurred character of such discourse, including the vague and often contradictory definitions and understandings of sunna circumcision, is well suited for negotiating changes in the challenging current landscape, where migrants must maneuver between traditional and international norms. As the term sunna circumcision offers both a sense of religious legitimacy and can be simultaneously perceived as a minor and extensive procedure, it can be acceptable to all groups, whether they adhere to traditional or international cultural norms.

## Supporting information

**S1 File. Interview guide.**
(DOCX)

**S2 File. Focus group guide.**
(DOCX)

**S3 File. Interview guide.**
(DOCX)

**S1 Fig. Flow chart of the study.** Adapted from Preferred Reporting Items for Systematic Reviews and Meta-Analyses Protocols (PRISMA-P). RCT = Randomized controlled trial.
(TIF)

## Acknowledgments

I want to thank our participants for their generosity in sharing their stories and insightful reflections. I am also most grateful for the diligence, engagement and reflections put into the work by my seven research assistants: Salma A. E. Ahmed, Khadra Yasien Ahmed, Naeema Saeed Sheekh Mohammed, Amira Jama Mohammed Ibrahim, Abdirizak Mohamud, Ibrahim Sheick Mohammed Ahmed and Omar Nur Gaal. Finally, I want to thank my colleague, Mai M. Ziyada for fruitful and inspiring discussions and support.

## Author Contributions

**Conceptualization:** R. Elise B. Johansen.

**Formal analysis:** R. Elise B. Johansen.

**Methodology:** R. Elise B. Johansen.

**Project administration:** R. Elise B. Johansen.

**Writing – original draft:** R. Elise B. Johansen.

**Writing – review & editing:** R. Elise B. Johansen.

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
