## [Decision Letter · Decision Letter 0]

18 Jan 2022

PONE-D-21-16363

Discourses of change from infibulation to sunna circumcision among Somali and Sudanese migrants in Norway

PLOS ONE

Dear Dr. Johansen,

Thank you for submitting your manuscript to PLOS ONE. After careful consideration, we feel that it has merit but does not fully meet PLOS ONE’s publication criteria as it currently stands. Therefore, we invite you to submit a revised version of the manuscript that addresses the points raised during the review process.

The manuscript has been evaluated by two reviewers, and their comments are available below.

The reviewers have provided an in-depth review of your manuscript and raised several major concerns which requires attention. In particular, they have noted that the authors have two related publications and requests clarification on how the study aim and conclusions reported in the current manuscript differ to that of the related articles. As mentioned previously, one of PLOS One’s publication criteria states "If a submitted study replicates or is very similar to previous work, authors must provide a sound scientific rationale for the submitted work and clearly reference and discuss the existing literature. Submissions that replicate or are derivative of existing work will likely be rejected if authors do not provide adequate justification." (https://journals.plos.org/plosone/s/criteria-for-publication#loc-2).

Furthermore, the reviewers have provided constructive feedback on focusing the scope of the discussion to the results shown.

Finally,  we feel that the manuscript will benefit from language copy editing. One of the publication criteria at PLOS ONE (https://journals.plos.org/plosone/s/criteria-for-publication#loc-5 )is that articles must be presented in an intelligible fashion and written in clear, correct, and unambiguous English. lease note that PLOS ONE cannot provide copyediting for manuscripts.

Could you please carefully revise the manuscript to address all comments raised?

We look forward to receiving your revised manuscript.

Kind regards,

Lucinda Shen, MSc

Staff Editor

PLOS ONE

Journal Requirements:

2. Please provide additional details regarding participant consent. In the ethics statement in the Methods and online submission information, please ensure that you have specified (i) whether consent was informed and (ii) what type you obtained (for instance, written or verbal, and if verbal, how it was documented and witnessed). If your study included minors, state whether you obtained consent from parents or guardians. If the need for consent was waived by the ethics committee, please include this information.

3. We noted in your submission details that a portion of your manuscript may have been presented or published elsewhere. "Three of the 25 quotes in this paper is presented in a former paper" Please clarify whether this  publication was peer-reviewed and formally published. If this work was previously peer-reviewed and published, in the cover letter please provide the reason that this work does not constitute dual publication and should be included in the current manuscript.

Reviewers' comments:

Reviewer's Responses to Questions

**Comments to the Author**

1. Is the manuscript technically sound, and do the data support the conclusions?

Reviewer #1: Partly

Reviewer #2: Yes

2. Has the statistical analysis been performed appropriately and rigorously? 

Reviewer #1: I Don't Know

Reviewer #2: N/A

3. Have the authors made all data underlying the findings in their manuscript fully available?

Reviewer #1: No

Reviewer #2: No

4. Is the manuscript presented in an intelligible fashion and written in standard English?

Reviewer #1: No

Reviewer #2: No

5. Review Comments to the Author

Reviewer #1: Overall

It is an interesting paper which attempts to explain the changing terminology and possible change in practices when it comes to FGM from the perspectives of the Sudanese and Somali diaspora living in Norway. The paper tries to unpick some of the terminology and perceptions of what those terms mean. This information is useful in understanding behaviour in order to make appropriate changes to public policy. However I do have some serious concerns.

I’m not sure about the validity of publishing a paper using results that have already been published in another 2 papers by the same author? Reference to paper 32 and paper 9. This needs to be better explained and dully justified. HAving read paper ref 9, the content is very similar and has already been published by PLOSONE.

The methodology is not clear with regards to other ‘informal conversations’ that are included. Neither is there a valid explanation of how the themes emerged or whether these were themes in fact created by the author. The results section needs to be better explained – as it is currently not clear which information is coming from which study of the two studies and which info is from FGDs and which are from in depth interviews. It is a very wordy paper for a scientific article – it would be more impressive if more succinct and to the point. The grammar is often not correct presumably because the author is not writing in his/her native language. I would suggest getting external support for this as I do not have the time to correct English grammar.

These are my specific points:

Intro

Quite the contrary do existing evidence suggest that the procedures performed in the name of sunna circumcision often differ only slightly from infibulation.

Is this a question?

To clarify the discourse, we will start by a brief outline

You need a full stop here.

WHO differentiates between four main types of FGC on the basis of anatomical extent and

severity, ranging from type IV with no tissue removal, to type I and II with tissue removal and Type III that also includes vaginal closure (11). Type IV includes practices such as pricking or nicking, Type I includes partial or total removal of the clitoral glans (the external and visible part of the clitoris), and/or the prepuce/ clitoral hood, Type II includes the partial or total removal of the clitoral glans and the labia minora, with or without removal of the labia 4 majora and Type III involves the narrowing of the vaginal opening through the creation of a covering seal formed by cutting and repositioning the labia minora, or labia majora, sometimes through stitching, with or without removal of the clitoral prepuce/clitoral hood and glans (11).

This is a very confusing description. It may be easier to put a table in listing 1 to IV with a description of each type alongside. If you donty want to add a table I suggest you at least describe tyhem in sequence – ie I to IV rather than moving between them randomly.

This meaning complex of infibulation

This does not make sense

Method and field

Maybe you mean ‘method and setting?’ or you could just use the word methodology.

This paper draws on findings from two recent studies exploring changes in the perception of

FGC among Somali and Sudanese migrants in Norway in 2017 to 2019 (9, 32).

I’m concerned about you using the data from two studies both of which have already been published to write a further paper putting the two together. You need to explain why you are covering new ground with this paper otherwise you are simply multiplying publications based on the same data.

Both studies also included participant observation and informal conversations with other Somali and Sudanese migrants. This included other Somali and Sudanese migrants in Norway, onward migrants, that is, Somali and Sudanese migrants who had left Norway to resettle in other countries of migration (UK and Canada), as well as return migrants, that is Somali and Sudanese migrants that had left Norway permanently to resettle in their countries of origin, including Sudan, Somalia and Kenya.

Its not clear who the participants in your study are? You mention the two earlier studies and then refer to informal conversations with other migrants – are they as well as the numbers in the other two studies. If so, how many more and what kind of qualitative methodology did you use. Informal conversations would not fit as a qualitative methodology.

Furthermore we had conversations with persons working in Non-Governmental Organizations (NGO) and government organizations dealing with FGC in a transnational context in Norway and Kenya, totaling more than 70 women and men.

What qualitative methodology did you use to get information from NGOs and the government? Were they formally part of your data – were they in depth interviews? Please be explicit about how they have been included in this research and what recognised qualitative methodology you used.

Research participants

Table 1

95 research participants

Do these include the NGO and government workers? They were not of Somali origin I presume? Please be explicit about who your research participants are, which are from your previous studies and which are extra to your previously published studies.

Thus, youth are defined as unmarried and young (under 30 years of age), whereas adults are defined as (ever been) married and age (above 24 years of age) (see table 1)

Your definition of youth and adults is confusing. You should be using the terms for what they are ie. ‘Never married’ and ‘married/or ever married’. Also, presumably you mean married or partnership as some may live together but never chose to get married?

Time of residence in Norway ranged from six months to thirty years, with an average of about 13.

You should be precise – not use terms like ’about.’ Please replace with the exact average.

Among the adults a little over half were employed

Please be exact.

Details about the study participants are deliberately vague or slightly altered

You cannot alter facts about the participants. This is a scientific journal and therefore details must be true and precise otherwise it makes your research invalid. Please find another way to protect the anonymity of your participants.

Reflexivity

I think you mean study limitations? Reflexivity is not a term that is used in scientific publications as far as I am aware.

Discourses and perceptions of change from infibulation to sunna type of

FGC

Most female study participants had undergone FGC, and most of them infibulation

Please be more precise. Exactly what perceptage had undergone FGC and what percentage had infibulation.

…we will work to carefully disentangle the two, and in in the following

outline each aspect of chance separately.

This sentence does not make sence.

Most commonly, study participants believed infibulation to have been imported from Egypt

Is it fair to say most commonly male study participants? If so, please add this.

Furthermore, was the transition from infibulation to sunna circumcision generally described in terms of FGC abandonment, suggesting that most participants did not consider sunna circumcision as a form of FGC

Please correct the English – it doesn’t make sense at the moment.

A midwife working at health center catering for Somali women in Nairobi taking part in an informal conversation, claimed that since FGC is done on small girls with equivalently small genitalia, two to three stitches would cause significant vaginal closure. Furthermore, she insisted that sunna circumcision nevertheless had to create a sufficiently small vaginal opening to satisfy the main purpose of FGC among Somalis, which is to create, protect and prove virginity. Thus, sunna circumcision had to include sufficient closure as to make sexual intercourse impossible without tearing or cutting parts of the infibulated seal.

If this is a quote from your informal interviews you have to reference it as so (eg Personal communications, date etc..) If you don’t reference it properly then you must explain this is not from the qualitative research but rather just an opinion of midwife working with Sudanese/Somalli women in Nairobi.

Sarah substantiated this description

Who is Sarah? Is this a mistake - including a name?

Discussion

we also found that the largest group expressed a sense of ambivalence, of being torn between traditional values and social expectations from family and network in country of origin adhering to traditional norms, and the international norms dominating their surroundings in country of migration

I am not aware that this aspect of international norms is addressed at all in this paper. Are you referring to the other paper published based on the second study? If so it seems strange to discuss it to such a large extent in the opening paragraph of your discussion. I would suggest you bring this idea in later, once you have discussed the findings of this current paper.

“I feel that a lot. Many times, I feel that they are cautious about their opinions in that

matter, for example some can say that circumcision is not existing, and they are against

it, but I feel that they are attached to their family’ traditions and believe that there is

something that has to be removed. For me, I know that there is a small thing to be

removed, but I don’t know the technique. But I feel that other people believe so too, but

they deny that. I see it in their faces that there is something, but they can’t admit, and

they say that they are against.

You cannot include a quote in your discussion. This should be added to your results section. I suggest you move it across and you can always refer to it in the discussion section.

My data also suggest that even those actively supporting sunna circumcision

would refrain from doing so due to the Norwegian ban on all types of FGC (55)

You mean ‘our data’ as you alone did not produce this I assume. Or replace by the name of the authors … et al.

However, while the aim of this paper is not to assess risk of FGC, the extent to which we have risk in mind, our focus is on how the ongoing change in meaning making accompanying a transition from infibulation to sunna circumcision may, our focus is more global.

Please correct the grammar

Though the data are collected in Norway, the explored discourse is ongoing globally, and we would not limit our concern to whether or not there is a risk of girls being cut while living in Norway, or whether Norwegian law is broken. We are also concerned about the risk of those 15% of Somalis with Norwegian citizenship that leave to settle in another country of migration or return to their country of origin (56). And beyond so-called holiday cutting, that our data suggest is rare, we are concerned about the about quarter of Somali children and teens in Europe that are sent to country of origin for so-called cultural rehabilitation (57). In such cases, where children commonly stay for a longer time (one to three years) under the custody of relatives in country of origin, there are anecdotal and documented cases where the locally responsible relatives arrange for FGC, with or without the girl’s parents’ knowledge or consents (32, 58, 59).

You are diverging quite a lot from the research discussed in your paper and the evidence it provides when discussing the risk of FGM to diaspora when they return to their country of origin. It is OK to make reference to how your findings might be relevant to policies as you do in the conclusion but you should not digress to an extensive elaboration of your opinions on this in the discussion.

Would have been interesting to look at how the discourse might be different amongst the participants who grew up in Norway or who had lived in Norway for a long time compared to those who had recently arrived. Do you have data on this – if so could you add?

Conclusion

Religiosity

I don’t think this is a word.

...was there several factors suggesting a less dramatic change than at first appearance.

Correct the grammar

Reviewer #2: The paper is important and interesting but could benefit from English proof reading. The data support conclusions but this could improve if the language was improved and clarifications made. The author have to my knowledge not been provided with the underlying data for the study.

6. PLOS authors have the option to publish the peer review history of their article (what does this mean?). If published, this will include your full peer review and any attached files.

Reviewer #1: No

Reviewer #2: No

---

## [Author Response · Author response to Decision Letter 0]

4 Mar 2022

Response to reviewers point by point - PONE-D-21-16363

Comments from the editor

a. Editors’ comment: Response to editor: There is no changes in financial disclosure.

b. Editors’ comment: Guidelines for resubmitting your figure files are available below the reviewer comments at the end of this letter.

Response to editor: A figure has been added (Figure 1, page 4), and uploaded accordingly.

c. Editors’ comment: Provide additional details regarding participant consent. - specified (i) whether consent was informed and (ii) type you obtained (for instance, written or verbal, and if verbal, how it was documented and witnessed). If your study included minors, state whether you obtained consent from parents or guardians. 

Response to editor: This information is now provide in the manuscript, in the method section, page 12, line 219-224.

d. Editors’ comment: We noted in your submission details that a portion of your manuscript may have been presented or published elsewhere. "Three of the 25 quotes in this paper is presented in a former paper" Please clarify whether this publication was peer-reviewed and formally published. If this work was previously peer-reviewed and published, in the cover letter please provide the reason that this work does not constitute dual publication and should be included in the current manuscript.

Response to editor: This is now clarified both in the cover letter to the editor* and the manuscript, page 10, line 188-197.

The following information is provided in the cover letter: Concerning the paper being based on data of which some is also used in other papers. Three out of the total of 32 quotes in the attached paper have been used in an earlier paper peer-reviewed and published by PlosOne (Johansen REB. Blurred transitions of female genital cutting in a Norwegian Somali community, 2019). However, this new paper does not constitute a dual publication for the following reasons: In contrast to the aforementioned paper that provides an overview of all the major themes found in the study, this paper goes in-depth into one of these themes to ensure a better empirical foundation and more nuances and further elaborations. For the current paper, the author re-read and re-analyzed all original data from both studies with to identify all data relevant for the discourse of change from infibulation to sunna circumcision, and a reanalysis of these data. Thus the new paper thus includes substantially more empirical data, with 29 additional quotes as well as additional empirical information in other forms. This to provide stronger empirical basis for the analysis and to provide more rich and nuanced data and analysis. Furthermore have the final analysis been expanded, including a further analysis of why the discourse of change seems to be both contradictory and blurred. Thus the discussion and conclusion does not only provide empirical data on what is happening, but also a further analysis of why this is taking place. 

e. Editors’ comment: In your Data Availability statement, you have not specified where the minimal data set underlying the results described in your manuscript can be found. PLOS defines a study's minimal data set as the underlying data used to reach the conclusions drawn in the manuscript and any additional data required to replicate the reported study findings in their entirety. All PLOS journals require that the minimal data set be made fully available. For more information about our data policy, please see http://journals.plos.org/plosone/s/data-availability.

Response to editor: Information on data-availability is now included in the cover letter. All relevant data is included in the manuscript.

f. Editors’ comment: Please include your full ethics statement in the ‘Methods’ section of your manuscript file. In your statement, please include the full name of the IRB or ethics committee who approved or waived your study, as well as whether or not you obtained informed written or verbal consent. If consent was waived for your study, please include this information in your statement as well. 

Response to editor: This is now done in method section, subsection on ethical considerations, page 11 lines 218 to page 12, 226.

Reviewer #1: Overall

1. Reviewer’s question/comment: It is an interesting paper which attempts to explain the changing terminology and possible change in practices when it comes to FGM from the perspectives of the Sudanese and Somali diaspora living in Norway. The paper tries to unpick some of the terminology and perceptions of what those terms mean. This information is useful in understanding behavior in order to make appropriate changes to public policy. However I do have some serious concerns.

I’m not sure about the validity of publishing a paper using results that have already been published in another 2 papers by the same author? Reference to paper 32 and paper 9. This needs to be better explained and dully justified. Having read paper ref 9, the content is very similar and has already been published by PLOSONE.

Response: I have now added an improved description on the difference between this and former papers from the same data-set on page 10, lines 188-197.

2. Reviewer’s question/comment: The methodology is not clear concerning other ‘informal conversations’ that are included. Neither is there a valid explanation of how the themes emerged or whether these were themes in fact created by the author. 

Response: I have now added a paragraph on the analyzing process, see page 10, lines 193-195. How the additional data (beyond the 95 study participants) was collected is outlined more detailed from page 9, line 162 to page 10, line 184. How these additional data are used are described on page 10, line 185-187.

3. Reviewer’s question/comment: The results section needs to be better explained – as it is currently not clear which information is coming from which study of the two studies and which info is from FGDs and which are from in depth interviews. 

Response: Regarding the quotes, in addition to gender and approximate age, II have now marked whether the statement came in an interview (IDI) or a focus group discussions (FGD). Quotes and stories from the second study are further marked with a *. Quotes and stories from the first study are unmarked. And, in the cases where a quote marked with a star is of a Somali person, this indicates that it comes from an interview conducted by the author. This is now explained in the method section, page 12, lines 236-242.

4. Reviewer’s question/comment: It is a very wordy paper for a scientific article – it would be more impressive if more succinct and to the point. 

Response: I have tried to shorten and sharpen the language, and cut or reduce some side-tracks (e.g. on clothing and social control, and my the concern of FGC after migration) to compensate for the extra words added by responding the reviewers requests for additional information, reducing the number of words substantially (from 11 438 to 10 282 words), now including acknowledgement. 

These are my specific points:

Introduction

5. Reviewer’s question/comment: Quite the contrary do existing evidence suggest that the procedures performed in the name of sunna circumcision often differ only slightly from infibulation. Is this a question? 

Response: Not a question, a statement. Grammar corrected to avoid misunderstanding. See page 3, line 37-38.

6. Reviewer’s question/comment: To clarify the discourse, we will start by a brief outline. You need a full stop here.

Response: Done. Sentence is slightly revised, page 3, line 45-46.

7. Reviewer’s question/comment: WHO differentiates between four main types of FGC on the basis of anatomical extent and severity, ranging from type IV with no tissue removal, to type I and II with tissue removal and Type III that also includes vaginal closure (11). Type IV includes practices such as pricking or nicking, Type I includes partial or total removal of the clitoral glans (the external and visible part of the clitoris), and/or the prepuce/ clitoral hood, Type II includes the partial or total removal of the clitoral glans and the labia minora, with or without removal of the labia 4 majora and Type III involves the narrowing of the vaginal opening through the creation of a covering seal formed by cutting and repositioning the labia minora, or labia majora, sometimes through stitching, with or without removal of the clitoral prepuce/clitoral hood and glans (11).

This is a very confusing description. It may be easier to put a table in listing 1 to IV with a description of each type alongside. If you donty want to add a table I suggest you at least describe tyhem in sequence – ie I to IV rather than moving between them randomly.

Response: I have now added a figure (Figure 1 page 4) and a table to clarify this (Table 1, page 4 and 5). The figure demonstrates what was formerly formulated in words in terms of the gradual increase in anatomical extent from type IV, followed by type I, II and III (Figure 1). This to clarify the common confusion made between numbers and severity, as the WHO typology describes increasingly severe procedures from I to III, whereas type IV is the least severe. In addition, I have added a table describing more detailed the similarities and differences between WHO classification and the emic classifications among Somalis and Sudanese at home and abroad in numerous order on the basis of the numbers in the WHO typology (Table 1). 

8. Reviewer’s question/comment: “This meaning complex of infibulation” -This does not make sense

Response: This text is altered for clarity, page 7, line 109-111.

9. Reviewer’s question/comment: Method and field. Maybe you mean ‘method and setting?’ or you could just use the word methodology.

Response: Changed now to “Materials and methods” in accordance with the PlosOne Guidelines. Page 7, line 124.

10. Reviewers’ question/comment: This paper draws on findings from two recent studies exploring changes in the perception of FGC among Somali and Sudanese migrants in Norway in 2017 to 2019 (9, 32). I’m concerned about you using the data from two studies both of which have already been published to write a further paper putting the two together. You need to explain why you are covering new ground with this paper otherwise you are simply multiplying publications based on the same data.

Response: This is now clarified both in the cover letter to the editor* and the manuscript, page 10, line 188-197. In the letter to the editor the following information is provided: Concerning the paper being based on studies that also are used in other publications. Three out of the current 32 quotes in the paper have been used in an earlier peer-reviewed and published paper in PlosOne (Johansen REB. Blurred transitions of female genital cutting in a Norwegian Somali community. 2019). However, this new paper does not constitute a dual publication for the following reasons: The two former paper presented an overview of the themes found, whereas the current paper provides an in-depth analysis of one of these themes. For the current paper, original data from the two studies were re-read and re-analyzed with a focus on identifying and analyzing all data relevant for the discourse of change from infibulation to sunna circumcision. The new paper thus includes substantially more empirical data, including 29 quotes not formerly published, as well new informant stories. The purpose was to provide a stronger empirical basis for the analysis and to provide more rich and nuanced analysis. Furthermore, I have expanded the final analysis with a stronger emphasis on the ways in which and the reasons why both discourse and practice appears blurred, i.e. to facilitate peoples maneuvering of change in a context of change, where they have to relate to both traditional expectations from countrymen, not least in country of origin where infibulation is still a norm, and the wider Norwegian and international society that judge FGC practices as human rights violation.

11. Reviewers’ question/comment: Both studies also included participant observation and informal conversations with other Somali and Sudanese migrants. This included other Somali and Sudanese migrants in Norway, onward migrants, that is, Somali and Sudanese migrants who had left Norway to resettle in other countries of migration (UK and Canada), as well as return migrants, that is Somali and Sudanese migrants that had left Norway permanently to resettle in their countries of origin, including Sudan, Somalia and Kenya. 

Its not clear who the participants in your study are? You mention the two earlier studies and then refer to informal conversations with other migrants – are they as well as the numbers in the other two studies. If so, how many more and what kind of qualitative methodology did you use. Informal conversations would not fit as a qualitative methodology.

Response: In this article I have used data from the two mentioned studies, which included formal interviews and focus group discussions with 95 Somali and Sudanese migrants. In addition I conducted “ethnographic fieldwork”, including interviews and participant observation in various settings. With one exemption, were none of the participants and data from the ethnographic fieldwork used directly in the paper or calculated as one of the 95 participants. Rather, these data were used to test the validity of the findings (member-checking), and provide contextual information to better inform the analysis. The one exception is the statement from a midwife interviewed during a stay in Kenya (now described in the method section (page 9, line 165-166), as suggested by editor in PlosOne. She is nevertheless not counted among the 95 study participants. This has now made clearer in the article (page 10 line 185-187). 

12. Reviewers’ question/comment: “Furthermore we had conversations with persons working in Non-Governmental Organizations (NGO) and government organizations dealing with FGC in a transnational context in Norway and Kenya, totaling more than 70 women and men. What qualitative methodology did you use to get information from NGOs and the government? Were they formally part of your data – were they in depth interviews? Please be explicit about how they have been included in this research and what recognized qualitative methodology you used.

Response: Se response to question above for clarification, and manuscript (page 10, line 185-187).

13. Reviewers’ question/comment: Research participants, Table 1, 95 research participants. Do these include the NGO and government workers? They were not of Somali origin I presume? Please be explicit about who your research participants are, which are from your previous studies and which are extra to your previously published studies.

Response: Interviews conducted as a part of my ethnographic fieldwork come in addition to the 95 study participants, including the NGO, though they were of Somali origin. Government workers are neither included in the 95 participants, and none of these were of Somali origin. No additional information was collected for the purpose of this new paper. However, the table with the overview of the informants have now been removed, and the information provided in the text (see page 8, line 146-148). 

14. Reviewers’ question/comment: “Thus, youth are defined as unmarried and young (under 30 years of age), whereas adults are defined as (ever been) married and age (above 24 years of age) (see table 1). Your definition of youth and adults is confusing. You should be using the terms for what they are ie. ‘Never married’ and ‘married/or ever married’. Also, presumably you mean married or partnership as some may live together but never chose to get married?

Response: I decided to remove the differentiation between youth and adults in the categorization of informants, but described how it was used to organize focus groups discussions and why, and how it was defined on page 8, line 140-148.

Some additional answers regarding marriage and co-habitation status. None of the participants claimed to be co-habiting without being married, probably because this would be strongly frowned upon as immoral within Somali and Sudanese network. On the other hand were some of the participants married, but not cohabitating, as the spousse lived in other towns or countries. These participants are probably registered as single in official systems in Norway. However, as this information did not seem to be relevant for the study findings, and would take space to discuss, I have not included it in this paper. 

15. Reviewers’ question/comment: Time of residence in Norway ranged from six months to thirty years, with an average of about 13. You should be precise – not use terms like ’about.’ Please replace with the exact average.

Response: As this is not a quantitative paper, and some participants did not provide information of their length of stay, I have removed the reference to an average – page 11, line 203-205.

16. Reviewers’ question/comment: Among the adults a little over half were employed. Please be exact.

Response: As this is a qualitative study, the data are not representative, nor did all participants provide such information, I do not find it useful to be so specific about actual numbers. However, I have now specified in the text that while we did not ask specifically about current employment or study status, most provided this. Page 11, lines 206-211.

17. Reviewers’ question/comment: Details about the study participants are deliberately vague or slightly altered. You cannot alter facts about the participants. This is a scientific journal and therefore details must be true and precise otherwise it makes your research invalid. Please find another way to protect the anonymity of your participants.

Response: I have now clarified in the text that in actually the data are not altered, only vague. And potential identifying data, such as age, number of children, marital status, or time of residence in Norway is not provided. See page 12, line 224-226.

18. Reviewers’ question/comment: Reflexivity. I think you mean study limitations? Reflexivity is not a term that is used in scientific publications as far as I am aware.

Response: In qualitative studies it has become increasingly common over the last 20 years to add a section on reflexivity. This is a response to an increased awareness of the role of the researcher in data-collection, and a perception of qualitative data as construed in collaboration between the researcher and the study participant, in which their similarities and differences in personal qualities, belonging, ethnicity and position, is likely to have an effect on access, trust and maybe also study findings (1, 2). It is even more important to include in studies of sensitive topics affecting a minority group. I have thus left the subtitle “reflexivity”. Thus in qualitative studies the position of researcher and data-collector is often elaborated in a section commonly entitled “reflexivity”, in contrasts to the more common reference to “study limitations” in quantitative studies. See for example: 

1. Ziyada MM, Lien IL, Johansen REB. Sexual norms and the intention to use healthcare services related to female genital cutting: A qualitative study among Somali and Sudanese women in Norway. PLOSE ONE. 2020.

2. Dodgson JE. Reflexivity in qualitative research. Journal of Human Lactation. 2019;35(2):220-2.

Also, the professional English language reviewers did not question the headline. 

19. Reviewers’ question/comment: Discourses and perceptions of change from infibulation to sunna type of- FGC. Most female study participants had undergone FGC, and most of them infibulation Please be more precise. Exactly what percentage had undergone FGC and what percentage had infibulation.

Response: As this is not a quantitative paper with representative sample, I do not consider it suitable to provide prevalence in percentages. Neither did we ask directly about FGC status, thus this was not always provided. However, I have however now given actual number of girls and women who claimed to have undergone infibulation, as well as those claiming sunna and no FGC., and those who gave no clear indication of type experienced. See page 11, line 212-216.

20. Reviewers’ question/comment: “…we will work to carefully disentangle the two, and in in the following outline each aspect of chance separately.2 This sentence does not make sence.

Response: Whole paragraph have been reworked for clarity.

21. Reviewers’ question/comment: “…Most commonly, study participants believed infibulation to have been imported from Egypt Is it fair to say most commonly male study participants? If so, please add this.

Response: The higher prevalence of this view, or more importantly how this made participants think about the practice of FGC as a foreign cultural element was more common among the men, and has now been highlighted in the article. Page 15, line 298-304.

22. Reviewers’ question/comment: “…Furthermore, was the transition from infibulation to sunna circumcision generally described in terms of FGC abandonment, suggesting that most participants did not consider sunna circumcision as a form of FGC Please correct the English – it doesn’t make sense at the moment.

Response: The sentence has been reworked for clarity, page 13, line 251-260.

23. Reviewers’ question/comment: “…A midwife working at health center catering for Somali women in Nairobi taking part in an informal conversation, claimed that since FGC is done on small girls with equivalently small genitalia, two to three stitches would cause significant vaginal closure. Furthermore, she insisted that sunna circumcision nevertheless had to create a sufficiently small vaginal opening to satisfy the main purpose of FGC among Somalis, which is to create, protect and prove virginity. Thus, sunna circumcision had to include sufficient closure as to make sexual intercourse impossible without tearing or cutting parts of the infibulated seal.

If this is a quote from your informal interviews you have to reference it as so (eg Personal communications, date etc..) If you don’t reference it properly then you must explain this is not from the qualitative research but rather just an opinion of midwife working with Sudanese/Somali women in Nairobi.

Response: This is from one of the informal interviews, yes. I have added the information on how and when this data was acquired as suggested by PlosOne, see page 9 line 165 to 167, page 10, line 185-187, and page 17, line 350-351

24. Reviewers’ question/comment: “…Sarah substantiated this description Who is Sarah? Is this a mistake - including a name?

Response: Pseudonym has now been removed and replaced with description of her as a study participant.

25. Reviewers’ question/comment: “…Discussion we also found that the largest group expressed a sense of ambivalence, of being torn between traditional values and social expectations from family and network in country of origin adhering to traditional norms, and the international norms dominating their surroundings in country of migration

I am not aware that this aspect of international norms is addressed at all in this paper. Are you referring to the other paper published based on the second study? If so it seems strange to discuss it to such a large extent in the opening paragraph of your discussion. I would suggest you bring this idea in later, once you have discussed the findings of this current paper.

Response: The suggestion is now followed, and the discussion section reworked. See page 26, line 588 onwards.

26. Reviewers’ question/comment: “… “I feel that a lot. Many times, I feel that they are cautious about their opinions in that matter, for example some can say that circumcision is not existing, and they are against it, but I feel that they are attached to their family’ traditions and believe that there is something that has to be removed. For me, I know that there is a small thing to be removed, but I don’t know the technique. But I feel that other people believe so too, but they deny that. I see it in their faces that there is something, but they can’t admit, and they say that they are against.

You cannot include a quote in your discussion. This should be added to your results section. I suggest you move it across and you can always refer to it in the discussion section.

Response: Quote is now removed from the paper as it did not fit into the finding section without expanding that extensively. 

27. Reviewers’ question/comment: “…My data also suggest that even those actively supporting sunna circumcision would refrain from doing so due to the Norwegian ban on all types of FGC (55) You mean ‘our data’ as you alone did not produce this I assume. Or replace by the name of the authors … et al.

Response: Done. Replaced with “our findings” to indicate how most data was collected by a team of research assistants. For more elaboration on research on this see page 8, lines 136-140 and page 9 lines 167-173.

28. Reviewers’ question/comment: “…However, while the aim of this paper is not to assess risk of FGC, the extent to which we have risk in mind, our focus is on how the ongoing change in meaning making accompanying a transition from infibulation to sunna circumcision may, our focus is more global. Please correct the grammar

Response: This sentence and discussion is now removed to shorten and sharpen the text as requested. 

29. Reviewers’ question/comment: “…Though the data are collected in Norway, the explored discourse is ongoing globally, and we would not limit our concern to whether or not there is a risk of girls being cut while living in Norway, or whether Norwegian law is broken. We are also concerned about the risk of those 15% of Somalis with Norwegian citizenship that leave to settle in another country of migration or return to their country of origin (56). And beyond so-called holiday cutting, that our data suggest is rare, we are concerned about the about quarter of Somali children and teens in Europe that are sent to country of origin for so-called cultural rehabilitation (57). In such cases, where children commonly stay for a longer time (one to three years) under the custody of relatives in country of origin, there are anecdotal and documented cases where the locally responsible relatives arrange for FGC, with or without the girl’s parents’ knowledge or consents (32, 58, 59).

You are diverging quite a lot from the research discussed in your paper and the evidence it provides when discussing the risk of FGM to diaspora when they return to their country of origin. It is OK to make reference to how your findings might be relevant to policies as you do in the conclusion but you should not digress to an extensive elaboration of your opinions on this in the discussion.

Response: This view is now taken into consideration and the text removed for the purpose of shortening and sharpening the manuscript.

30. Reviewers’ question/comment: “…Would have been interesting to look at how the discourse might be different amongst the participants who grew up in Norway or who had lived in Norway for a long time compared to those who had recently arrived. Do you have data on this – if so could you add?

Response: Time of residence (in terms of actual years or approximate or real age at arrival) was provided by most, but not all participants. However, there was no systematic differences in perceptions and views in terms of length of stay. Some participants had been engaged in activities against FGC already in country of origin, whereas some had not thought about the practice even after 10 – 15 years in Norway. Hence, contrary to what has been suggested in quantitative studies, this study does not indicate systematic differences between length of stay and insight and opinion is thus not discussed in the paper.

31. Reviewers’ question/comment: “…Conclusion Religiosity I don’t think this is a word.

Response: The term exist and is used according to google, google translate and English dictionaries. Furthermore was the word not questioned by the professional proof readers.

32. Reviewers’ question/comment: “…...was there several factors suggesting a less dramatic change than at first appearance. Correct the grammar

Response: Done. See page 3, line 36-37.

Reviewer #2: 

I. Reviewers’ question/comment: “…The paper is important and interesting but could benefit from English proof reading. The data support conclusions but this could improve if the language was improved and clarifications made. 

Response: The paper has been sent for English proof reading. As this is expensive, I did this only after receiving and responding to the reviews. 

II. Reviewers’ question/comment: “…The author have to my knowledge not been provided with the underlying data for the study.

Response: I am not sure what is meant by this comment, but yes, the author have conducted a number of the interviews personally, and received transcripts of all the other interviews and focus group discussions, hence provided with all underlying data for the paper. This is now clarified in the method section, see page 8, line 148-153. 

III. Reviewers’ question/comment: Very important and interesting paper with informative content and interesting points brought to light. However, the paper would benefit from undergoing English proof reading, sometimes the meaning gets lost due to incorrect (I assume) use of words. 

Response: Proofreading is conducted after reception of the reviews and my subsequent revision of manuscript.

IV. Reviewers’ question/comment: Also, you sometimes write ‘we’ and sometimes ‘me’. 

Response: I have written this paper alone, whereas most of the data was collected by research assistants. I have double checked the manuscript to ensure that I, me and we are used correctly.

Introduction: 

V. Reviewers’ question/comment: You have two aims, it would be clearer if you place these together, perhaps turn them into a broader aim and two objectives.

Response: This is now done, page 3, line 39-44. 

The result section is clear and informative.

Discussion:

VI. Reviewers’ question/comment: The discussion is a bit messy and could benefit from clarification, could do with subheadings. I suggest you try to separate the points a bit, perhaps discuss sexuality by itself, and then tradition/religion, and then ‘modern/backwards’. I think this will provide some clarification. Also look over how you use ‘real’ and ‘ideal’ and ‘discourse’ and ‘reality’, and how these connect to each other.

Interesting discussion on religion here.

Response: Have reworked the discussion section with these commentaries in mind, though not adding subheadings, the text is not reorganized as suggested. See page 26, line 587 onwards.

Conclusion:

VII. Reviewers’ question/comment: I like your conclusion as your points, but not sure if the point about sexual control to move from outer (infibulation) to inner (sunna) works – except for in the discourse (if the ‘real’ sunna is closure anyway). Perhaps one point that could be highlighted more is that without the changes in meaning of female sexuality (ie that it still needs to be controlled), a change to lesser forms of FGC (or abandonment) is unlikely to take place, at least in more than the discourse. This does come across, but could be clearer.

Response: Both the findings, the discussion and conclusion have been revised to clarify the suggested points that change from infibulation to sunna circumcision is mainly a discursive change. However, I do not think that it is impossible to change to a less extensive sunna without changing the meaning of female sexuality, as I have now stronger highlighted that sexual control is also increasingly thought to be possible through other outer means than cutting, including clothing and increased social control. However, due to space I have only been able to allude to this. Rather I have strengthened the discussion and conclusion on how the blurred aspects of the discourse facilitates maneuvering in the current context of change. 

Specific comments:

VIII. Reviewers’ question/comment: Page 3, first half: Please provide a reference to the sentence: Nevertheless, existing evidence suggest that the change of type is both less widespread and substantial than commonly claimed. 

Response: References are now added, page 3, line 37.

IX. Reviewers’ question/comment: Look over this sentence, Furthermore, the change of anatomical extent of FGC is associated with changes in the meaning making, in which religion seems to play a major role, while concerns over female sexuality, a major component of infibulation, still is a major cultural and religious value. Can it be divided and clarified?

Response: The whole paragraph is now change, page 27, line 588 onwards. See alse page 29, line 641-644.

Page 4:

X. Reviewers’ question/comment: Thus, the contrast between infibulation and sunna circumcision can, in some cases, be limited to a slight difference in the extent of vaginal closure. Perhaps specify that you mean the discursive contrast here.

Response: The whole text is reworked to be more specific and illustrated in a figure (Figure 1, page 4) and a table (Table 1, page 4-5). I have also reworked the text to highlight that the difference can be slight both discursively and in practice.

Page 7: Methods and field

XI. Reviewers’ question/comment: Do you mean that you draw on data from two studies that also resulted in two other publications? Or do you only draw on the result/findings from these two published studies? Please clarify here. 

Response: I draw on data from two studies that also resulted in two other papers. This is now made clearer in the text, page 10, line 188-197.

XII. Reviewers’ question/comment: Also, did you include the conversations with NGO officials in your analysis? Or did these merely interpret your analysis? This could also need clarification.

Response: Findings from discussions with NGO officials contributed to my overall understanding of the discourse. This is now clarified in the method section.(page 10, lines 185-187).

XIII. Reviewers’ question/comment: Also, could you include some of the questions (or perhaps topic guide) used for the interviews? Were they all the same or did they differ?

Response: I have now included a paragraph on the topics explored in the interviews and focus group discussions in the two studies, see page 9, line 154-161.

Page 8:

XIV. Reviewers’ question/comment: What does it add to the study to divide the groups into ‘youth’ and ‘adults’? I find it a bit confusing, also because some can go into both groups and because it is different in men. I understand that this can be an interesting point for analysis and discussion, but do not see the purposefulness in the division on this level. Also, what about those women aged 23 in the table? Even if there were no such women. Please look over the table and make it clearer. 

Response: I have now removed the differentiation between youth and adults, expect for its role in the composition of the focus groups and why and how it was defined, see page 8, line 140-147. As the result section is not divided by age group, I have also removed the table on this, but added this info in the text, page 8, line 146-148.

Page 9:

XV. Reviewers’ question/comment: Regarding education, perhaps avoid using the word ‘ significantly’ as this implies a quantitative terminology and measurement.

Response: I have removed the term significant to avoid misunderstandings, page 11, line 211.

Ethical considerations: 

XVI. Reviewers’ question/comment: Do you mean that you changed some of the characteristics when providing the quotes? Or in the table? Please specify.

Response: This is now specified, in the sense that that the I did not alter any information about the individual participants in this final manuscript, just providing limited or vague data (such as approximate age, without providing information about marital status, educational level or time of residence for each participant), page 11, line 224-226.

Page 10:

XVII. Reviewers’ question/comment: Reflexivity: in the first paragraph, I think you can reflect a little bit more about how being an insider/outsider might have influenced the responses, or at least why/how you can to the conclusion that this did not affect your data. The second paragraph of the reflexivity chapter I do not feel belong here but earlier in the methods chapter. 

Response: I have added a reflections on this and its lack of any visible difference in this study, and kept the reference to a more elaborately discussed of this in another paper that was based on the first study, page 1, lines 228-235 + 338-242. Second paragraph is now reworked and the information moved further up front. 

XVIII. Reviewers’ question/comment: Please specify where you start indicating your findings.

Response: This is now done, page 12, line 243. 

XIX. Reviewers’ question/comment: Page 11: the wording ‘The flow of the discourse’ is used a bit lightly here and it is not clear what this means. I suggest a different, more specific, wording here. Also, how do this quote indicate that this means a change towards abandonment as opposed to change from one FGC to another? Could you provide some more evidence to support this suggestion? 

Response: This text have now been strongly altered for clarification of these questions and one example added on page 13, line 251-260.

XX. Reviewers’ question/comment: Page 14: It would be great if you had a quote to support the 40-year old women who considered her extensive ‘sunna’ to be minor and harmless.

Response: As this information had to be deducted from different statements throughout the interview and thus be difficult to illustrate by one or two quotes, I have omitted this story from the manuscript, but added more broad discussions and additional data on the participants’ conceptualization of harm caused by sunna in the section on sunna, page 15 line 313 onwards. 

XXI. Reviewers’ question/comment: Page 15: Interesting quote by this woman. It would be good, however, to get a little bit more information about why she knows how the ‘sunna’ looks in many women – is this something the majority of women know (how it is in others) because they talk to each other or does she have special insight (being a doctor/midwife or something)?

Response: I have now added a couple of sentences of this, page 18, line 370-374.

XXII. Reviewers’ question/comment: Also, can you give a bit more detail on how ‘sunna’ is proscribed by religion? Perhaps in the background?

Response: Some additional information on this is now provide in the introduction on page 7, lines 112-120. 

XXIII. Reviewers’ question/comment: Page 29: What is the difference between the papers building up to the study, to this study? What can you draw on from that paper, and what comes (in result) of the present paper? Did you have different conclusions in that paper than in this one? 

Response: Yes, the other papers building on the data presented an overview of the different themes that were recurrent and dominant in the findings, whereas this paper focus in-depth on one of these themes. This is now clearer expressed on page 10, line 188-197. Furthermore is the conclusion different, in that it focuses more on the blurredness of the discourse as a way to maneuver in times of contradictory social norms and change.

---

## [Editor Report · Decision Letter 1]

28 Apr 2022

Discourses of change: The shift from infibulation to sunna circumcision among Somali and Sudanese migrants in Norway

PONE-D-21-16363R1

Dear Dr. Johansen,

We’re pleased to inform you that your manuscript has been judged scientifically suitable for publication and will be formally accepted for publication once it meets all outstanding technical requirements.

Kind regards,

Malin Jordal, Ph.D.

Guest Editor

PLOS ONE

Additional Editor Comments (optional):

Congratulation with a clear and well-written paper which highlights the complexities involved in the transition from infibulation to sunna circumcision in Somali and Sudanese populations in diaspora. The paper contributes to a nuanced discussion on the subject of FGC and change.

I discovered a few small unclarities and/or spelling mistakes, indicated below. Please address these before submitting your final version.

Line 180: it should be ‘additional’ here.

Line 212: write ‘n’ instead of ‘no’ in the bracket.

Line 238-239: this line is a little bit unclear.

Line 313: should the first word be ‘when’ (not ‘while’)?

Line 314: this reference to ‘well-described practice’ reads a bit awkward here, I cannot see that the following indicates that it is not ‘well-defined’. Is faraonic ‘well-defined’ and sunna not? The following sentence reads well though, perhaps just remove the first part of the sentence?

Line 241-242: this is very interesting, but a little bit clear. Can you specify what the woman meant by the writing end and the rubber end of a pen and how this related to the two types (infibulation and sunna)? Perhaps indicating approximately centimeters?

Line 372: remove the word ‘thus’?

Line 448: spelling mistake, it should be ‘it’.

Line 669-671: there seems to be a grammatical error in this sentence.
---

## [Editor Report · Acceptance letter]

26 May 2022

PONE-D-21-16363R1 

DISCOURSES OF CHANGE: THE SHIFT FROM INFIBULATION TO SUNNA CIRCUMCISION AMONG SOMALI AND SUDANESE MIGRANTS IN NORWAY 

Dear Dr. Johansen:

I'm pleased to inform you that your manuscript has been deemed suitable for publication in PLOS ONE. Congratulations! Your manuscript is now with our production department. 

Kind regards, 

on behalf of

Dr. Malin Jordal 

Guest Editor

PLOS ONE